# Bayesian optimization for automated model selection

**Gustavo Malkomes,**[†] **Chip Schaff,**[†] **Roman Garnett**
Department of Computer Science and Engineering
Washington University in St. Louis
St. Louis, MO 63130
{luizgustavo, cbschaff, garnett}@wustl.edu

## Abstract

Despite the success of kernel-based nonparametric methods, kernel selection still requires considerable expertise, and is often described as a "black art." We present a sophisticated method for automatically searching for an appropriate kernel from an infinite space of potential choices. Previous efforts in this direction have focused on traversing a kernel grammar, only examining the data via computation of marginal likelihood. Our proposed search method is based on Bayesian optimization in model space, where we reason about model evidence as a function to be maximized. We explicitly reason about the data distribution and how it induces similarity between potential model choices in terms of the explanations they can offer for observed data. In this light, we construct a novel kernel between models to explain a given dataset. Our method is capable of finding a model that explains a given dataset well without any human assistance, often with fewer computations of model evidence than previous approaches, a claim we demonstrate empirically.

## 1 Introduction

Over the past decades, enormous human effort has been devoted to machine learning; preprocessing data, model selection, and hyperparameter optimization are some examples of critical and often expert-dependent tasks. The complexity of these tasks has in some cases relegated them to the realm of "black art." In kernel methods in particular, the selection of an appropriate kernel to explain a given dataset is critical to success in terms of the fidelity of predictions, but the vast space of potential kernels renders the problem nontrivial. We consider the problem of automatically finding an appropriate probabilistic model to explain a given dataset. Although our proposed algorithm is general, we will focus on the case where a model can be completely specified by a kernel, as is the case for example for centered Gaussian processes (GPs).

Recent work has begun to tackle the kernel-selection problem in a systematic way. Duvenaud et al. [1] and Grosse et al. [2] described generative grammars for enumerating a countably infinite space of arbitrarily complex kernels via exploiting the closure of kernels under additive and multiplicative composition. We adopt this kernel grammar in this work as well. Given a dataset, Duvenaud et al. [1] proposed searching this infinite space of models using a greedy search mechanism. Beginning at the root of the grammar, we traverse the tree greedily attempting to maximize the (approximate) evidence for the data given by a GP model incorporating the kernel.

In this work, we develop a more sophisticated mechanism for searching through this space. The greedy search described above only considers a given dataset by querying a model's evidence. Our search performs a *metalearning* procedure, which, conditional on a dataset, establishes similarities among the models in terms of the space of explanations they can offer for the data. With this viewpoint, we construct a novel kernel between models (a "kernel kernel"). We then approach

---

[†]These authors contributed equally to this work

the model-search problem via Bayesian optimization, treating the model evidence as an expensive black-box function to be optimized as a function of the kernel. The dependence of our kernel between models on the distribution of the data is critical; depending on a given dataset, the kernels generated by a compositional grammar could be especially rich or deceptively so.

We develop an automatic framework for exploring a set of potential models, seeking the model that best explains a given dataset. Although we focus on Gaussian process models defined by a grammar, our method could be easily extended to any probabilistic model with a parametric or structured model space. Our search appears to perform competitively with other baselines across a variety of datasets, including the greedy method from [1], especially in terms of the number of models for which we must compute the (expensive) evidence, which typically scales cubically for kernel methods.

## 2    Related work

There are several works attempting to create more expressive kernels, either by combining kernels or designing custom ones. *Multiple kernel learning* approaches, for instance, construct a kernel for a given dataset through a weighted sum of predefined and fixed set of kernels, adjusting the weights to best explain the observed data. Besides limiting the space of kernels considered, the hyperparameters of component kernels also need to be specified in advance [3, 4]. Another approach to is to design flexible kernel families [5–7]. These methods often use Bochner's theorem to reason in spectral space, and can approximate any arbitrary stationary kernel function. In contrast, our method does not depend on stationarity. Other work has developed expressive kernels by combining Gaussian processes with deep belief networks; see, for example, [8–10]. Unfortunately, there is no free lunch; these methods require complicated inference techniques that are much more costly than using standard kernels.

The goal of *automated machine learning* (autoML) is to automate complex machine-learning procedures using insights and techniques from other areas of machine learning. Our work falls into this broad category of research. By applying machine learning methods throughout the entire modeling process, it is possible to create more automated and, eventually, better systems. Bergstra et al. [11] and Snoek et al. [12], for instance, have shown how to use modern optimization tools such as Bayesian optimization to set the hyperparameters of machine learning methods (e.g., deep neural networks and structured SVMs). Our approach to model search is also based on Bayesian optimization, and its success in similar settings is encouraging for our adoption here. Gardner et al. [13] also considered the automated model selection problem, but in an *active leaning* framework with a fixed set of models. We note that our method could be adopted to their Bayesian active model selection framework with minor changes, but we focus on the classical supervised learning case with a fixed training set.

## 3    Bayesian optimization for model search

Suppose we face a classical supervised learning problem defined on an input space $\mathcal{X}$ and output space $\mathcal{Y}$. We are given a set of training observations $\mathcal{D} = (\mathbf{X}, \mathbf{y})$, where $\mathbf{X}$ represents the design matrix of explanatory variables $\mathbf{x}_i \in \mathcal{X}$, and $y_i \in \mathcal{Y}$ is the respective value or label to be predicted. Ultimately, we want to use $\mathcal{D}$ to predict the value $y_*$ associated with an unseen point $\mathbf{x}_*$. Given a probabilistic model $\mathcal{M}$, we may accomplish this via formation of the predictive distribution.

Suppose, however, that we are given a collection of probabilistic models $\mathbb{M}$ that could have plausibly generated the data. Ideally, finding the source of $\mathcal{D}$ would let us solve our prediction task with the highest fidelity. Let $\mathcal{M} \in \mathbb{M}$ be a probabilistic model, and let $\Theta_{\mathcal{M}}$ be the corresponding parameter space. These models are typically parametric families of distributions, each of which encodes a *structural* assumption about the data, for example, that the data can be described by a linear, quadratic, or periodic trend. Further, the member distributions ($\mathcal{M}_\theta \in \mathcal{M}$, $\theta \in \Theta_{\mathcal{M}}$) of $\mathcal{M}$ differ from each other by a particular value of some properties—represented by the *hyperprameters* $\theta$—related to the data such as amplitude, characteristic length scales, etc.

We wish to select one model from this collection of models $\mathbb{M}$ to explain $\mathcal{D}$. From a Bayesian perspective, the principle approach for solving this problem is *Bayesian model selection*.[2] The critical

value is *model evidence*, the probability of generating the observed data given a model $\mathcal{M}$:

$$p(\mathbf{y} \mid \mathbf{X}, \mathcal{M}) = \int_{\Theta_\mathcal{M}} p(\mathbf{y} \mid \mathbf{X}, \theta, \mathcal{M}) \, p(\theta \mid \mathcal{M}) \, \mathrm{d}\theta. \tag{1}$$

The evidence (also called *marginal likelihood*) integrates over $\theta$ to account for all possible explanations of the data offered by the model, under a prior $p(\theta \mid \mathcal{M})$ associated with that model.

Our goal is to automatically explore a space of models $\mathbb{M}$ to select a model[3] $\mathcal{M}^* \in \mathbb{M}$ that explains a given dataset $\mathcal{D}$ as well as possible, according to the model evidence. The essence of our method, which we call *Bayesian optimization for model search* (BOMS), is viewing the evidence as a function $g \colon \mathbb{M} \to \mathbb{R}$ to be optimized. We note two important aspects of $g$. First, for large datasets and/or complex models, $g$ is an expensive function, for example growing cubically with $|\mathcal{D}|$ for GP models. Further, gradient information about $g$ is impossible to compute due to the discrete nature of $\mathbb{M}$. We can, however, query a model's evidence as a black-box function. For these reasons, we propose to optimize evidence over $\mathbb{M}$ using *Bayesian optimization,* a technique well-suited for optimizing expensive, gradient-free, black-box objectives [14]. In this framework, we seek an optimal model

$$\mathcal{M}^* = \arg\max_{\mathcal{M} \in \mathbb{M}} g(\mathcal{M}; \mathcal{D}), \tag{2}$$

where $g(\mathcal{M}; \mathcal{D})$ is the (log) model evidence:

$$g(\mathcal{M}; \mathcal{D}) = \log p(\mathbf{y} \mid \mathbf{X}, \mathcal{M}). \tag{3}$$

We begin by placing a Gaussian process (GP) prior on $g$,

$$p(g) = \mathcal{GP}(g; \mu_g, K_g),$$

where $\mu_g \colon \mathbb{M} \to \mathbb{R}$ is a mean function and $K_g \colon \mathbb{M}^2 \to \mathbb{R}$ is a covariance function appropriately defined over the model space $\mathbb{M}$. This is a nontrivial task due to the discrete and potentially complex nature of $\mathbb{M}$. We will suggest useful choices for $\mu_g$ and $K_g$ when $\mathbb{M}$ is a space of Gaussian process models below. Now, given observations of the evidence of a selected set of models,

$$\mathcal{D}_g = \left\{ \left( \mathcal{M}_i, g(\mathcal{M}_i; \mathcal{D}) \right) \right\}, \tag{4}$$

we may compute the posterior distribution on $g$ conditioned on $\mathcal{D}_g$, which will be an updated Gaussian process [15]. Bayesian optimization uses this probabilistic belief about $g$ to induce an inexpensive acquisition function to select which model we should select to evaluate next. Here we use the classical *expected improvement* (EI) [16] acquisition function, or a slight variation described below, because it naturally considers the trade off between exploration and exploitation. The exact choice of acquisition function, however, is not critical to our proposal. In each round of our model search, we will evaluate the acquisition function in the optimal model evidence for a number of candidate models $\mathcal{C}(\mathcal{D}_g) = \{\mathcal{M}_i\}$, and compute the evidence of the candidate where this is maximized:

$$\mathcal{M}' = \arg\max_{\mathcal{M} \in \mathcal{C}} \alpha_{\mathrm{EI}}(\mathcal{M}; \mathcal{D}_g).$$

We then incorporate the chosen model $\mathcal{M}'$ and the observed model evidence $g(\mathcal{M}'; \mathcal{D})$ into our model evidence training set $\mathcal{D}_g$, update the posterior on $g$, select a new set of candidates, and continue. We repeat this iterative procedure until a budget is expended, typically measured in terms of the number of models considered.

We have observed that expected improvement [16] works well especially for small and/or low-dimensional problems. When the dataset is large and/or high-dimensional, training costs can be considerable and variable, especially for complex models. To give better anytime performance on such datasets, we use *expected improvement per second,* where we divide the expected improvement by an estimate of the time required to compute the evidence. In our experiments, this estimation was performed by fitting a linear regression model to the log time to compute $g(\mathcal{M}; \mathcal{D})$ as a function of the number of hyperparameters (the dimension of $\Theta_\mathcal{M}$) that we train on the models available in $\mathcal{D}_g$.

The acquisition function allows us to quickly determine which models are more promising than others, given the evidence we have observed so far. Since $\mathbb{M}$ is an infinite set of models, we cannot consider every model in every round. Instead, we will define a heuristic to evaluate the acquisition function at a smaller set of active candidate models below.

# 4 Bayesian optimization for Gaussian process kernel search

We introduced above a general framework for searching over a space of probabilistic models $\mathbb{M}$ to explain a dataset $\mathcal{D}$ without making further assumptions about the nature of the models. In the following, we will provide specific suggestions in the case that all members of $\mathbb{M}$ are Gaussian process priors on a latent function.

We assume that our observations $\mathbf{y}$ were generated according to an unknown function $f\colon \mathcal{X} \to \mathbb{R}$ via a fixed probabilistic observation mechanism $p(\mathbf{y} \mid \mathbf{f})$, where $f_i = f(\mathbf{x}_i)$. In our experiments here, we will consider regression with additive Gaussian observation noise, but this is not integral to our approach. We further assume a GP prior distribution on $f$, $p(f) = \mathcal{GP}(f; \mu_f, K_f)$, where $\mu_f\colon \mathcal{X} \to \mathbb{R}$ is a mean function and $K_f\colon \mathcal{X}^2 \to \mathbb{R}$ is a positive-definite covariance function or kernel. For simplicity, we will assume that the prior on $f$ is centered, $\mu_f(x) = 0$, which lets us fully define the prior on $f$ by the kernel function $K_f$. We assume that the kernel function is parameterized by hyperparameters that we concatenate into a vector $\theta$. In this restricted context, a model $\mathcal{M}$ is completely determined by the choice of kernel function and an associated hyperparameter prior $p(\theta \mid \mathcal{M})$. Below we briefly review a previously suggested method for constructing an infinite space of potential kernels to model the latent function $f$, and thus an infinite family of models $\mathbb{M}$. We will the discuss the standardized and automated construction of associated hyperparameter priors.

## 4.1 Space of compositional Gaussian processes kernels

We adopt the same space of kernels defined by Duvenaud et al. [1], which we briefly summarize here. We refer the reader to the original paper for more details. Given a set of simple, so-called *base kernels,* such as the common squared exponential (SE), periodic (PER), linear (LIN), and rational quadratic (RQ) kernels, we create new and potentially complex kernels by summation and multiplication of these base units. The entire kernel space can be describe by the following grammar rules:

1. Any subexpression $\mathcal{S}$ can be replaced with $\mathcal{S} + \mathcal{B}$, where $\mathcal{B}$ is a base kernel.
2. Any subexpression $\mathcal{S}$ can be replaced with $\mathcal{S} \times \mathcal{B}$, where $\mathcal{B}$ is a base kernel.
3. Any base kernel $\mathcal{B}$ may be replaced with another base kernel $\mathcal{B}'$.

## 4.2 Creating hyperparameter priors

The base kernels we will use are well understood, as are their hyperparameters, which have simple interpretations that can be thematically grouped together. We take advantage of the Bayesian framework to encode prior knowledge over hyperparameters, i.e., $p(\theta \mid \mathcal{M})$. Conveniently, these priors can also potentially mitigate numerical problems during the training of the GPs. Here we derive a consistent method to construct such priors for arbitrary kernels and datasets in regression problems.

We first standardize the dataset, i.e., we subtract the mean and divide by the standard deviation of both the predictive features $\{x_i\}$ and the outputs $\mathbf{y}$. This gives each dataset a consistent scale. Now we can reason about what real-world datasets usually look like in this scale. For example, we do not typically expect to see datasets spanning $10\,000$ length scales. Here we encode what we judge to be reasonable priors for groups of thematically related hyperparameters for most datasets. These include three types of hyperparameters common to virtually any problem: length scales $\ell$ (including, for example, the period parameter of a periodic covariance), signal variance $\sigma$, and observation noise $\sigma_n$. We also consider separately three other parameters specific to particular covariances we use here: the $\alpha$ parameter of the rational quadratic covariance [15, (4.19)], the "length scale" of the periodic covariance $\ell_p$ [15, $\ell$ in (4.31)], and the offset $\sigma_0$ in the linear covariance. We define the following:

$$p(\log \ell) = \mathcal{N}(0.1, 0.7^2) \qquad p(\log \sigma) = \mathcal{N}(0.4, 0.7^2) \qquad p(\log \sigma_n) = \mathcal{N}(0.1, 1^2)$$
$$p(\log \alpha) = \mathcal{N}(0.05, 0.7^2) \qquad p(\log \ell_p) = \mathcal{N}(2, 0.7^2) \qquad p(\sigma_0) = \mathcal{N}(0, 2^2)$$

Given these, each model was given an independent prior over each of its hyperparameters, using the appropriate selection from the above for each.

## 4.3 Approximating the model evidence

The model evidence $p(\mathbf{y} \mid \mathbf{X}, \mathcal{M})$ is in general intractable for GPs [17, 15]. Alternatively we use a *Laplace approximation* to approximately compute the model evidence. This approximation works by

making a second-order Taylor expansion of $\log p(\theta \mid \mathcal{D}, \mathcal{M})$ around its mode $\hat{\theta}$ and approximates the model evidence as follows:

$$\log p(\mathbf{y} \mid \mathbf{X}, \mathcal{M}) \approx \log p(\mathbf{y} \mid \mathbf{X}, \hat{\theta}, \mathcal{M}) + \log p(\hat{\theta} \mid \mathcal{M}) - \tfrac{1}{2} \log \det \Sigma^{-1} + \tfrac{d}{2} \log 2\pi, \quad (5)$$

where $d$ is the dimension of $\theta$ and $\Sigma^{-1} = -\nabla^2 \log p(\theta \mid \mathcal{D}, \mathcal{M})\big|_{\theta=\hat{\theta}}$ [18, 19]. We can view (5) as rewarding model fit while penalizing model complexity. Note that the *Bayesian information criterion* (BIC), commonly used for model selection and also used by Duvenaud et al. [1], can be seen as an approximation to the Laplace approximation [20, 21].

## 4.4 Creating a "kernel kernel"

In §4.1, §4.2, and §4.3, we focused on modeling a latent function $f$ with a GP, creating an infinite space of models $\mathbb{M}$ to explain $f$ (along with associated hyperparameter priors), and approximating the log model evidence function $g(\mathcal{M}; \mathcal{D})$. The evidence function $g$ is the objective function we are trying to optimize via Bayesian optimization. We described in §3 how this search progresses in the general case, described in terms of an arbitrary Gaussian process prior on $g$. Here we will provide specific suggestions for the modeling of $g$ in the case that the model family $\mathbb{M}$ comprises Gaussian process priors on a latent function $f$, as discussed here and considered in our experiments.

Our prior belief about $g$ is given by a GP prior $p(g) = \mathcal{GP}(g; \mu_g, K_g)$, which is fully specified by the mean $\mu_g$ and covariance functions $K_g$. We define the former as a simple constant mean function $\mu_g(\mathcal{M}) = \theta_\mu$, where $\theta_\mu$ is a hyperparameter to be learned through a regular GP training procedure given a set of observations. The latter we will construct as follows.

The basic idea in our construction is that is that we will consider the distribution of the observation locations in our dataset $\mathcal{D}$, $\mathbf{X}$ (the design matrix of the underlying problem). We note that selecting a model class $\mathcal{M}$ induces a prior distribution over the latent function values at $\mathbf{X}$, $p(\mathbf{f} \mid \mathbf{X}, \mathcal{M})$:

$$p(\mathbf{f} \mid \mathbf{X}, \mathcal{M}) = \int p(\mathbf{f} \mid \mathbf{X}, \mathcal{M}, \theta) \, p(\theta \mid \mathcal{M}) \, \mathrm{d}\theta.$$

This prior distribution is an infinite mixture of multivariate Gaussian prior distributions, each conditioned on specific hyperparameters $\theta$. We consider these prior distributions as different explanations of the latent function $f$, restricted to the observed locations, offered by the model $\mathcal{M}$. We will compare two models in $\mathbb{M}$ according to how different the explanations they offer for $\mathbf{f}$ are, *a priori*.

The *Hellinger distance* is a probability metric that we adopt as a basic measure of similarity between two distributions. Although this quantity is defined between arbitrary probability distributions (and thus could be used with non-GP model spaces), we focus on the multivariate normal case. Suppose that $\mathcal{M}, \mathcal{M}' \in \mathbb{M}$ are two models that we wish to compare, in the context of explaining a fixed dataset $\mathcal{D}$. For now, suppose that we have conditioned each of these models on arbitrary hyperparameters (that is, we select a particular prior for $f$ from each of these two families), giving $\mathcal{M}_\theta$ and $\mathcal{M}'_{\theta'}$, with $\theta \in \Theta_\mathcal{M}$ and $\theta' \in \Theta_{\mathcal{M}'}$. Now, we define the two distributions

$$P = p(\mathbf{f} \mid \mathbf{X}, \mathcal{M}, \theta) = \mathcal{N}(\mathbf{f}; \mu_P, \Sigma_P) \qquad Q = p(\mathbf{f} \mid \mathbf{X}, \mathcal{M}', \theta') = \mathcal{N}(\mathbf{f}; \mu_Q, \Sigma_Q).$$

The squared *Hellinger distance* between $P$ and $Q$ is

$$d_{\mathrm{H}}^2(P, Q) = 1 - \frac{|\Sigma_P|^{1/4} |\Sigma_Q|^{1/4}}{\left| \frac{\Sigma_P + \Sigma_Q}{2} \right|^{1/2}} \exp\left\{ -\frac{1}{8} (\mu_P - \mu_Q)^\top \left( \frac{\Sigma_P + \Sigma_Q}{2} \right)^{-1} (\mu_P - \mu_Q) \right\}. \quad (6)$$

The Hellinger distance will be small when $P$ and $Q$ are highly overlapping, and thus $\mathcal{M}_\theta$ and $\mathcal{M}'_{\theta'}$ provide similar explanations *for this dataset*. The distance will be larger, conversely, when $\mathcal{M}_\theta$ and $\mathcal{M}'_{\theta'}$ provide divergent explanations. Critically, we note that this distance depends on the dataset under consideration in addition to the GP priors.

Observe that the distance above is not sufficient to compare the similarity of two models $\mathcal{M}, \mathcal{M}'$ due to the fixing of hyperparameters above. To properly account for the different hyperparameters of different models, and the priors associated with them, we define the *expected squared Hellinger distance* of two models $\mathcal{M}, \mathcal{M}' \in \mathbb{M}$ as

$$\bar{d}_{\mathrm{H}}^2(\mathcal{M}, \mathcal{M}'; \mathbf{X}) = \mathbb{E}\left[ d_{\mathrm{H}}^2(\mathcal{M}_\theta, \mathcal{M}'_{\theta'}) \right] = \iint d_{\mathrm{H}}^2(\mathcal{M}_\theta, \mathcal{M}'_{\theta'}; \mathbf{X}) \, p(\theta \mid \mathcal{M}) \, p(\theta' \mid \mathcal{M}') \, \mathrm{d}\theta \, \mathrm{d}\theta', \quad (7)$$

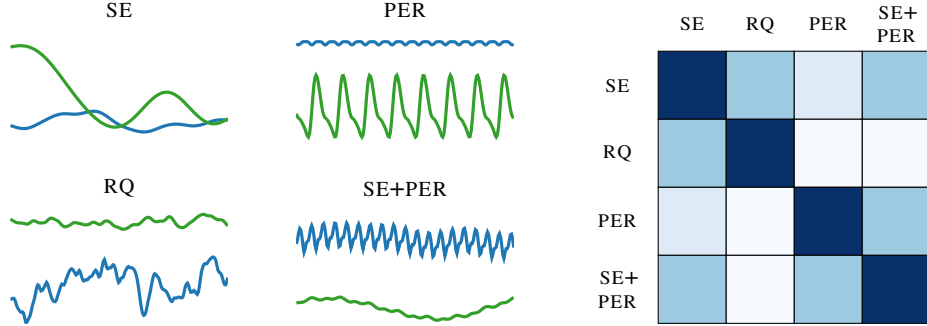

Figure 1: A demonstration of our model kernel $K_g$ (8) based on expected Hellinger distance of induced latent priors. Left: four simple model classes on a $1d$ domain, showing samples from the prior $p(f \mid \mathcal{M}) \propto p(f \mid \theta, \mathcal{M}) \, p(\theta \mid \mathcal{M})$. Right: our Hellinger squared exponential covariance evaluated for the grid domains on the left. Increasing intensity indicates stronger covariance. The sets $\{\text{SE}, \text{RQ}\}$ and $\{\text{SE}, \text{PER}, \text{SE+PER}\}$ show strong mutual correlation.

where the distance is understood to be evaluated between the priors provided on $\mathbf{f}$ induced at $\mathbf{X}$. Finally, we construct the *Hellinger squared exponential* covariance between models as

$$K_g(\mathcal{M}, \mathcal{M}'; \theta_g, \mathbf{X}) = \sigma^2 \exp\left(-\frac{1}{2}\frac{\bar{d}_{\text{H}}^2(\mathcal{M}, \mathcal{M}'; \mathbf{X})}{\ell^2}\right), \tag{8}$$

where $\theta_g = (\sigma, \ell)$ specifies output and length scale hyperparameters in this kernel/evidence space. This covariance is illustrated in Figure 1 for a few simple kernels on a fictitious domain.

We make two notes before continuing. The first observation is that computing (6) scales cubically with $|\mathbf{X}|$, so it might appear that we might as well compute the evidence instead. This is misleading for two reasons. First, the (approximate) computation of a given model's evidence via either a Laplace approximation or the BIC requires optimizing its hyperparameters. Especially for complex models this can require hundreds-to-thousands of computations that each require cubic time. Further, as a result of our investigations, we have concluded that in practice we may approximate (6) and (7) by considering only a small *subset* of the observation locations $\mathbf{X}$ and that this usually sufficient to capture the similarity between models in terms of explaining a given dataset. In our experiments, we choose 20 points uniformly at random from those available in each dataset, fixed once for the entire procedure and for all kernels under consideration in the search. We then used these points to compute distances (6–8), significantly reducing the overall time to compute $K_g$.

Second, we note that the expectation in (7) is intractable. Here we approximate the expectation via quasi-Monte Carlo, using a low-discrepancy sequence (a Sobol sequence) of the appropriate dimension, and inverse transform sampling, to give consistent, representative samples from the hyperparameter space of each model. Here we used 100 $(\theta, \theta')$ samples with good results.

## 4.5   Active set of candidate models

Another challenging of exploring an infinite set of models is how we advance the search. Each round, we only compute the acquisition function on a set of candidate models $\mathcal{C}$. Here we discuss our policy for creating and maintaining this set. From the kernel grammar (§4.1), we can define a model graph where two models are connected if we can apply one rule to produce the other. We seek to traverse this graph, balancing exploration (diversity) against exploitation (models likely to have higher evidence). We begin each round with a set of already chosen candidates $\mathcal{C}$. To encourage exploitation, we add to $\mathcal{C}$ all neighbors of the best model seen thus far. To encourage exploration, we perform random walks to create diverse models, which we also add to $\mathcal{C}$. We start each random walk from the empty kernel and repeatedly apply a random number of grammatical transformations. The number of such steps is sampled from a geometric distribution with termination probability $\frac{1}{3}$. We find that 15 random walks works well. To constrain the number of candidates, we discard the models with the lowest EI values at the end of each round, keeping $|\mathcal{C}|$ no larger than 600.

Table 1: Root mean square error for model-evidence regression experiment.

| Dataset | Train % | Mean | $k$-NN (SP) | $k$-NN ($\bar{d}_H$) | GP ($\bar{d}_H$) |
|---|---|---|---|---|---|
| CONCRETE | 20 | 0.109 (0.000) | 0.200 (0.020) | 0.233 (0.008) | **0.107** (0.001) |
| | 40 | 0.107 (0.000) | 0.260 (0.025) | 0.221 (0.007) | **0.102** (0.001) |
| | 60 | 0.107 (0.000) | 0.266 (0.007) | 0.215 (0.005) | **0.097** (0.001) |
| | 80 | 0.106 (0.000) | 0.339 (0.015) | 0.200 (0.003) | **0.093** (0.002) |
| HOUSING | 20 | 0.210 (0.001) | 0.226 (0.002) | 0.347 (0.004) | **0.175** (0.002) |
| | 40 | 0.207 (0.001) | 0.235 (0.004) | 0.348 (0.004) | **0.140** (0.002) |
| | 60 | 0.206 (0.000) | 0.235 (0.004) | 0.348 (0.004) | **0.123** (0.002) |
| | 80 | 0.206 (0.000) | 0.257 (0.004) | 0.344 (0.004) | **0.114** (0.002) |
| MAUNA LOA | 20 | 0.543 (0.002) | 0.736 (0.051) | 0.685 (0.010) | **0.513** (0.003) |
| | 40 | 0.537 (0.001) | 0.878 (0.062) | 0.667 (0.005) | **0.499** (0.003) |
| | 60 | 0.535 (0.001) | 1.051 (0.058) | 0.686 (0.010) | **0.487** (0.004) |
| | 80 | 0.534 (0.001) | 1.207 (0.048) | 0.707 (0.005) | **0.474** (0.004) |

## 5 Experiments

Here we evaluate our proposed algorithm. We split our evaluation into two parts: first, we show that our GP model for predicting a model's evidence is suitable; we then demonstrate that our model search method quickly finds a good model for a range of regression datasets. The datasets we consider are publicly available[4] and were used in previous related work [1, 3]. AIRLINE, MAUNA LOA, METHANE, and SOLAR are $1d$ time series, and CONCRETE and HOUSING have, respectively, 8 and 13 dimensions. To facilitate comparison of evidence across datasets, we report log evidence divided by dataset size, redefining

$$g(\mathcal{M}; \mathcal{D}) = \log(p(\mathbf{y} \mid \mathbf{X}, \mathcal{M}))/|\mathcal{D}|. \qquad (9)$$

We use the aforementioned base kernels $\{\text{SE}, \text{RQ}, \text{LIN}, \text{PER}\}$ when the dataset is one-dimensional. For multi-dimensional datasets, we consider the set $\{\text{SE}_i\} \cup \{\text{RQ}_i\}$, where the subscript indicates that the kernel is applied only to the $i$th dimension. This setup is the same as in [1].

### 5.1 Predicting a model's evidence

We first demonstrate that our proposed regression model in model space (i.e., the GP on $g\colon \mathbb{M} \to \mathbb{R}$) is sound. We set up a simple prediction task where we predict model evidence on a set of models given training data. We construct a dataset $\mathcal{D}_g$ (4) of 1 000 models as follows. We initialize a set $\mathbb{M}$ with the set of base kernels, which varies for each dataset (see above). Then, we select one model uniformly at random from $\mathbb{M}$ and add its neighbors in the model grammar to $\mathbb{M}$. We repeat this procedure until $|\mathbb{M}| = 1\,000$ and computed $g(\mathcal{M}; \mathcal{D})$ for the entire set generated. We train several baselines on a subset of $\mathcal{D}_g$ and test their ability to predict the evidence of the remaining models, as measured by the root mean squared error (RMSE). To achieve reliable results we repeat this experiment ten times. We considered a subset of the datasets (including both high-dimensional problems), because training 1 000 models demands considerable time. We compare with several alternatives:

1. **Mean prediction.** Predicts the mean evidence on the training models.
2. $k$-**nearest neighbors.** We perform $k$-NN regression with two distances: shortest-path distance in the directed model graph described in §4.5 (SP), and the expected squared Hellinger distance (7). Inverse distance was used as weights.

We select $k$ for both $k$-NN algorithms through cross-validation, trying all values of $k$ from 1 to 10. We show the average RMSE along with standard error in Table 1. The GP with our Hellinger distance model covariance universally achieves the lowest error. Both $k$-NN methods are outperformed by the simple mean prediction. We note that in these experiments, many models perform similarly in terms of evidence (usually, this is because many models are "bad" in the same way, e.g., explaining the dataset away entirely as independent noise). We note, however, that the GP model is able to exploit correlations in *deviations* from the mean, for example in "good pockets" of model space, to achieve

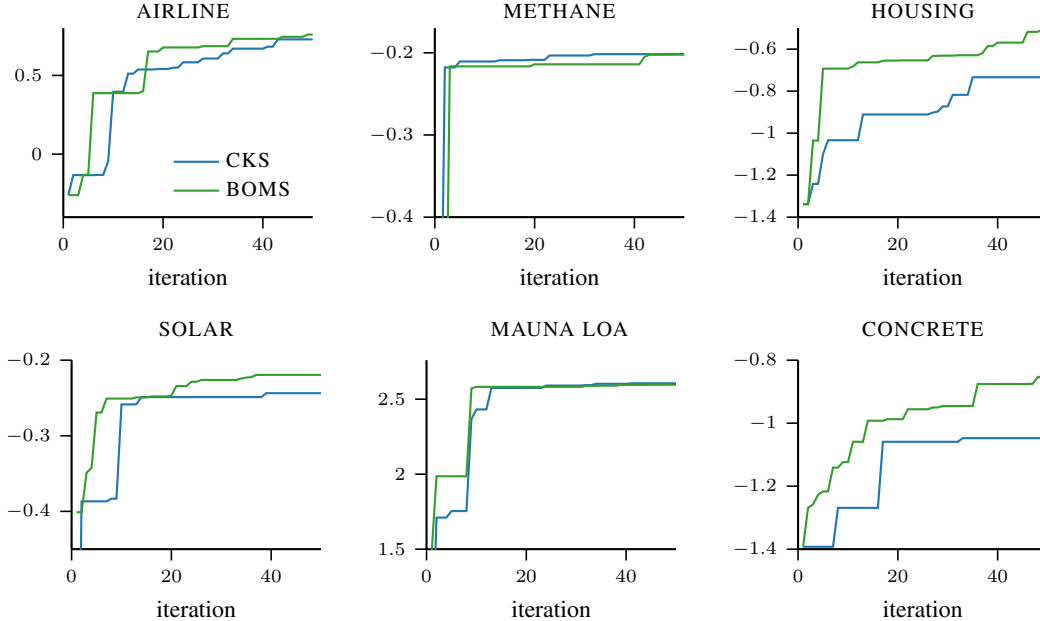

Figure 2: A plot of the best model evidence found (normalized by $|\mathcal{D}|$, (9)) as a function of the number of models evaluated, $g(\mathcal{M}^*; \mathcal{D})$, for six of the datasets considered (identical vertical axis labels omitted for greater horizontal resolution).

better performance. We also note that both the $k$-NN and GP models have decreasing error with the number of training models, suggesting our novel model distance is also useful in itself.

## 5.2 Model search

We also evaluate our method's ability to quickly find a suitable model to explain a given dataset. We compare our approach with the greedy *compositional kernel search* (CKS) of [1]. Both algorithms used the same kernel grammar (§4.1), hyperparameter priors (§4.2), and evidence approximation (§4.3, (5)). We used L-BFGS to optimize model hyperparameters, using multiple restarts to avoid bad local maxima; each restart begins from a sample from $p(\theta \mid \mathcal{M})$.

For BOMS, we always began our search evaluating SE first. The active set of models $\mathcal{C}$ (§4.5) was initialized with all models that are at most two edges distant from the base kernels. To avoid unnecessary re-training over $g$, we optimized the hyperparameters of $\mu_g$ and $K_g$ every 10 iterations. This also allows us to perform rank-one updates for fast inference during the intervening iterations. Results are depicted in Figure 2 for a budget of 50 evaluations of the model evidence. In four of the six datasets we substantially outperform CKS. Note the vertical axis is in the log domain. The overhead for computing the kernel $K_g$ and performing the inference about $g$ was approximately 10% of the total running time. On MAUNA LOA our method is competitive since we find a model with similar quality, but earlier. The results for METHANE, on the other hand, indicate that our search initially focused on a suboptimal region of the graph, but we eventually do catch up.

## 6 Conclusion

We introduced a novel automated search for an appropriate kernel to explain a given dataset. Our mechanism explores a space of infinite candidate kernels and quickly and effectively selects a promising model. Focusing on the case where the models represent structural assumptions in GPs, we introduced a novel "kernel kernel" to capture the similarity in prior explanations that two models ascribe to a given dataset. We have empirically demonstrated that our choice of modeling the evidence (or marginal likelihood) with a GP in model space is capable of predicting the evidence value of unseen models with enough fidelity to effectively explore model space via Bayesian optimization.

**Acknowledgments**

This material is based upon work supported by the National Science Foundation (NSF) under award number IIA−1355406. Additionally, GM acknowledges support from the Brazilian Federal Agency for Support and Evaluation of Graduate Education (CAPES).

## Footnotes

[2]"Model selection" is unfortunately sometimes also used in GP literature for the process of hyperparameter learning (selecting some $\mathcal{M}_\theta \in \mathcal{M}$), rather than selecting a model class $\mathcal{M}$, the focus of our work.

[3] We could also select a *set* of models but, for simplicity, we assume that there is one model that best explains that data with overwhelming probability, which would imply that there is not benefit in considering more than one model, e.g., via Bayesian model averaging.

[4] https://archive.ics.uci.edu/ml/datasets.html

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
