[Reviews · NeurIPS 2016]

Reviewer 1

Summary

This paper describes a Bayesian optimization technique that can be used in the context of finding a good kernel to explain the observed data. For this, a kernel over kernels is proposed. This allows to use probabilistic methods, such as Gaussian processes, to estimate the marginal likelihood of a particular kernel. The predictive distribution provided by the Gaussian process is exploited to explore the space of kernels more efficiently. In particular, it allows to use typical Bayesian optimization techniques to find a kernel with good performance on the datasets of interest. The proposed approach is evaluated on several datasets and compared with other methods previously used for that purpose, based on a greedy search.

Qualitative Assessment

The paper is very well written and it addresses a problem that is of interested to the machine learning community. The paper also comments on important related work and states the differences and advantages of the proposed method. The proposed technique to build a kernel over kernel is also interesting and may have other applications different from the one described in this paper. The experiments described in the paper, although not exhaustive, are enough to illustrate the good performance of the proposed approach. Something that can be criticized is the fact that evaluating the acquisition function looks expensive, since it has a cubic cost with respect to the number of observed samples. The authors comment that evaluating the marginal likelihood is even more expensive due to the use of the Laplace approximation. However, it is not clear how much expensive is that. In particular, the authors report in Figure 2 results with respect to the number of model's evaluations. Given that the application considered in the paper is very specific, it would have been better to report marginal likelihood with respect to the computational time.

Confidence in this Review

2-Confident (read it all; understood it all reasonably well)


Reviewer 2

Summary

The paper proposes an automatic method to search for a good kernel that can explain the data. They do this by treating the model evidence as an expensive black-box function that needs to be optimized as a function of the kernel function. The kernel function is specified using a compositinal grammer of base kernel functions. Optimization is done by casting it as a Bayesian optimization problem.

Qualitative Assessment

The paper presents an interesting automatic approach using Bayesian optimization for automatically finding an appropriate kernel function that can explain the data. This seems to be a good and quite logical extension of automating the use of compositional grammers of base kernel functions. The paper is well written and not technically difficult. Bayesian optimisation has been around quite some time and is well known to many. The main contribution of the paper is to formulate kernel search as a Bayesian optimisation problem and make reasonable choices for it to work. The authors clearly formulate the problem and clearly motivate the choices made in their framework (e.g., grammer, hyperparameter priors, approximations, kernel-of-kernel, etc.) to make it work. There are probably plenty of other options one could choose and argument for, but the choices made seem reasonable and described well. One thing that might be included is a reference to the work on hyperkernels.

Confidence in this Review

2-Confident (read it all; understood it all reasonably well)


Reviewer 3

Summary

This paper introduces a novel approach to kernel choice, especially in the case of Gaussian processes, through Bayesian optimization. A very interesting kernel between kernels is proposed. While the paper focuses on GPs, the method could be used more generally, and the techniques developed (approximating the model evidence, a "kernel kernel") are potentially useful beyond this setting.

Qualitative Assessment

I think the clarity of your presentation can be strengthened. 1) Line 80 - hyperprameters -> hyperparameters 2) Line 66 - adopted -> adapter 3) Line 83 - "principle approach" -> principled approach 4) Footnote 1 - maybe I'm one of the authors who's to blame, but I'm perfectly comfortable thinking of model selection as hyperparameter learning. Sure, I'd like to integrate out the hyperparameters (and thus average over a class of models), but if I can't do that, then I'm happy to learn them and select a single model. Indeed, I'd question your claim that the Bayesian perspective is one of Bayesian model selection---you're just moving one more level up the hierarchy (hyper-hyper parameters!), but wouldn't it be even better if we could integrate over different models M? This might actually be possible within your framework---you're using Bayesian optimization to maximize a function, but the recent probabilistic numerics work on Bayesian quadrature suggests you could instead use BO to do integration! 5) I was a little confused about how you maximize the acquisition function---where does the set of candidate models C come from, and what exactly do you have to do for each candidate model. I think run an optimization to approximate the model evidence, is that right? If so, it's confusing me because the model evidence approximation isn't discussed until later -- page 5 while the acquisition function is discussed on page 3. 6) The priors you suggest are reasonable, though I wonder if it'd be better to use a distribution like Student t (see some work by Vehtari and colleagues) in place of a Normal. Do you have any citations to back up your choices? Why sd = 0.7? 7) I'd like a little more discussion in Section 4.3. Would you suggest using Laplace in place of the use of BIC in Duvenaud et al? You should provide a page number in Murphy, as I'm curious to know more about BIC vs Laplace. 8) Section 4.4 is very interesting. Eq. 6 looks like what happens when you create a cross-covariance of two SE kernels (e.g. Melkumyan and Ramos, IJCAI 2011) though of course the setup is different. I'm also curious whether if there's an alternative approach to creating a kernel kernel by considering the maximum mean discrepancy between the distributions. 9) After rereading Section 5.1, I'm still a little confused about what you're evaluating. You should more clearly say what your method is, and why/what the mean prediction / k-nearest neighbors regression are doing. I think what you've left implicit is exactly what the ground truth is to which you're comparing. Once you say that, and say what your method is doing, everything should become clear.

Confidence in this Review

2-Confident (read it all; understood it all reasonably well)


Reviewer 4

Summary

The paper introduces a novel method of automated model selection which is driven by the maximization of the model evidence over a grammar of models. The basic idea is to use Bayesian global optimization to sequentially query the model space. This requires building a regression map between models and their evidence. The authors achieve this by building a covariance function that quantifies the similarity between two given models by comparing the Hellinger distance of prior distributions they induce over the targets. The definition of this covariance function constitutes one of the major contributions of this work. Another contribution of the paper is the on the fly construction of the active set of models, i.e., the set of models over which they look for the maximum expected improvement in the evidence, to promote exploration and exploitation. The authors test their methodology on six standard datasets and compare their proposed methodology to compositional kernel search (CKS) with positive results in both sets of tests.

Qualitative Assessment

This is a well written paper. The topic is important and there are at least two novel contributions (covariance over model space, and the generation of the active models for promoting exploration and exploitation). The application is very important. Making Bayesian model selection routinely available for a wide range of machine learning techniques is an extremely important quest. I presume that the field will gradually start moving towards the application of Bayesian global optimization techniques for tackling this problem. The present paper is an important contribution towards this goal. Please consider the following comments: 1. I know that there is not enough space in nine pages, but if you were able to show what are the kernel structures that you discover for one or two of your example datasets, that would be great. Being able to visualize the predictions would also be great. See if this can be squeezed in somehow. 2. A possible extension of the methodology would be the following. Instead of using the Laplace approximation for the model evidence, you could make use a noisy estimate of it obtained, e.g., by sequential Monte Carlo. By adding a nugget to your kernel kernel you will be able to filter this noise out on the fly. There are alternative acquisition function that can deal with noisy objectives (e.g., extended expected improvement or the knowledge gradient). This would extend your methodology to models in which the Laplace approximation fails.

Confidence in this Review

3-Expert (read the paper in detail, know the area, quite certain of my opinion)


Reviewer 5

Summary

The authors propose a Bayesian optimization framework to learn model evidence and use it as a criterion for model selection. For the case of Gaussian Process models, the paper shows a "kernel kernel" based on Hellinger distance to measure the similarity of GPs. Integrals involved are computed via Laplace approximation or quasi-Monte Carlo methods. Experiments show that the proposed algorithm is able to learn the model evidence and can choose better model compared to CKS.

Qualitative Assessment

The main novelty of this paper is a new Bayesian optimization algorithm for choosing Gaussian Process models. Though the authors indicate the framework is general, I am not sure what the performance would be in other cases since it involves several complex integrals which can only be computed via approximations. I am concerned about the experiment part and have several questions: - By Mean Prediction do you mean simply average the model evidence of the training set and predict this constant on the test set? If so, it is a rather naive algorithm and I am surprised that it outperforms both k-NN algorithms in Table. 1. Do you have any intuitive explanations about this phenomenon? Also, does it mean that the models are uniformly bad and it is very difficult to predict? Similarly, we observe that as the size of training dataset increases, k-NN (SP) performs even worse, which is rather bizarre. - I notice that the authors choose 20 points to evaluate the Hellinger distance. It would be interesting to see experiments analyzing the effect of increasing or decreasing this number. - In Lines 116-120 the authors propose to use expected improvement per second. I expect to see running time comparisons for each dataset and algorithm, since the number of evaluations of model evidence may not reflect the time consumed. - In the model search experiment the authors compare their algorithm to CKS on several small datasets, but is CKS the state-of-the-art? It is also interesting to see whether BOMS can improve the results of manually tuned models published before.

Confidence in this Review

2-Confident (read it all; understood it all reasonably well)


Reviewer 6

Summary

Summary: The paper presents an approach to automatically and efficiently select a kernel for GP regression from a certain infinite set. The approach models the likelihoods for each kernel as a GP and for that purpose the authors present a kernel for kernels using an approximation to the Hellinger distance.

Qualitative Assessment

Technical Quality: 3/5 There is not much to complain about the experiments and their analysis. Using only 20 datapoints (line 225) independent of the dataset seems a bit arbitrary. For a 4 star rating please provide (appendix) experiments showing how well or crude the approximation is. Novelty: 3/5 The idea to use the Hellinger distance is neat. Impact: 3/5 Selecting an appropriate kernel is a difficult task and certainly hinders many potential users from using GPs. This work does alleviate this problems but without open and easy-to-use software the impact is likely to be small. Clarity: 4/5 The paper is well written and easy to read. Minor issues: * please add a citation to the Hellinger distance (between Gaussians) * line 190: two times "is that" * line 233: Probably you mean "Another challenge in ..."?

Confidence in this Review

2-Confident (read it all; understood it all reasonably well)